# Polynomial-Time Optimal Equilibria with a Mediator in Extensive-Form Games

**Brian Hu Zhang**
Computer Science Department
Carnegie Mellon University
`bhzhang@cs.cmu.edu`

**Tuomas Sandholm**
Computer Science Department, CMU
Strategic Machine, Inc.
Strategy Robot, Inc.
Optimized Markets, Inc.
`sandholm@cs.cmu.edu`

## Abstract

For common notions of correlated equilibrium in extensive-form games, computing an optimal (*e.g.*, welfare-maximizing) equilibrium is NP-hard. Other equilibrium notions—*communication* [11] and *certification* [12] *equilibria*—augment the game with a mediator that has the power to both send and receive messages to and from the players—and, in particular, to remember the messages. In this paper, we investigate both notions in extensive-form games from a computational lens. We show that optimal equilibria in both notions can be computed in polynomial time, the latter under a natural additional assumption known in the literature. Our proof works by constructing a *mediator-augmented game* of polynomial size that explicitly represents the mediator's decisions and actions. Our framework allows us to define an entire family of equilibria by varying the mediator's information partition, the players' ability to lie, and the players' ability to deviate. From this perspective, we show that other notions of equilibrium, such as extensive-form correlated equilibrium, correspond to the mediator having *imperfect recall*. This shows that, at least among all these equilibrium notions, the hardness of computation is driven by the mediator's imperfect recall. As special cases of our general construction, we recover 1) the polynomial-time algorithm of Conitzer and Sandholm [8] for *automated mechanism design* in Bayes-Nash equilibria and 2) the *correlation DAG* algorithm of Zhang et al. [31] for optimal correlation. Our algorithm is especially scalable when the equilibrium notion is what we define as the *full-certification* equilibrium, where players cannot lie about their information but they can be silent. We back up our theoretical claims with experiments on a suite of standard benchmark games.

## 1 Introduction

Various equilibrium notions in general-sum extensive-form games are used to describe situations where the players have access to a trusted third-party *mediator*, who can communicate with the players. Depending on the power of the mediator and the form of communication, these notions include the *normal-form* [1] and *extensive-form correlated equilibrium* (NFCE and EFCE) [29], the *normal-form* [25] and *extensive-form* [10] *coarse-correlated equilibrium* (NFCCE and EFCCE), the *communication equilibrium* [11], and the *certification equilibrium* [12].

Several of these notions, in particular the EFCE and EFCCE, were defined for mainly *computational* reasons: the EFCE as a computationally-reasonable relaxation to NFCE, and the EFCCE as a computationally-reasonable relaxation of EFCE. When the goal is to compute a *single* correlated equilibrium, these relaxations are helpful: there are polynomial-time algorithms for computing an EFCE [16]. However, from the perspective of computing *optimal* equilibria—that is, equilibria that

36th Conference on Neural Information Processing Systems (NeurIPS 2022).

maximize the expected value of a given function, such as the social welfare—even these relaxations fall short: for all of the *correlation* notions above, computing an optimal equilibrium of an extensive-form game is NP-hard [29, 10].

On the other hand, notions of equilibrium involving *communication* in games have arisen. These differ from the notions of *correlation* in that the mediator can receive and remember information from the players, and therefore pass information *between* players as necessary to back up their suggestions. *Certification equilibria* [12] further strengthen communication equilibria by allowing players to *prove* certain information to the mediator. To our knowledge, the computational complexity of optimal communication or certification equilibria has never been studied. We do so in this paper. The main technical result of our paper is a *polynomial-time algorithm* for computing optimal communication and certification equilibria (the latter under a certain natural condition about what messages the players can send). This stands in stark contrast to the notions of correlation discussed above.

To prove our main result, we define a general class of *mediator-augmented games*, each having polynomial size, that is sufficient to describe all of the above notions of equilibrium except the NFCE[1]. We also build on this main result in several ways.

1. We define the *full-certification* equilibrium, which is the special case in which players cannot lie to the mediator (but can opt out of revealing their information). In this case, the algorithm is a linear program whose size is *almost linear* in the size of the original game. As such, this special case scales extremely well compared to all of the other notions.

2. We formalize notions for incorporating *payments* in the language of our augmented game. By using payments, mediators can incentivize players to play differently than they otherwise would, possibly to the benefit of the mediator's utility function.

3. We define an entire family of equilibria using our augmented game, that includes as special cases the communication equilibrium, certification equilibrium, NFCCE, EFCCE, and EFCE. From this perspective, we show that other notions of equilibrium, such as extensive-form correlated equilibrium, correspond to the mediator having *imperfect recall*. This shows that, at least among all these equilibrium notions, the hardness of computation is driven by the mediator's imperfect recall. We argue that, for this reason, many stated practical applications of correlated equilibria should actually be using communication or certification equilibria instead, which are both easier to compute (in theory, at least) and better at modelling the decision-making process of a rational mediator.

4. We empirically verify the above claims via experiments on a standard set of game instances.

**Applications and related work.** Correlated and communication equilibria have various applications that have been well-documented. Here, we discuss just a few of them, as motivation for our paper. For further discussion of related work, especially relating to automated dynamic mechanism design and persuasion, see Appendix F.

*Bargaining, negotiation, and conflict resolution* [4, 9]. Two parties with asymmetric information wish to arrive at an agreement, say, the price of an item. A mediator, such as a central third-party marketplace, does not know the players' information but can communicate with the players.

*Crowdsourcing and ridesharing* [13, 22, 31]. A group of players each has individual goals (*e.g.*, to make money by serving customers at specific locations). The players are coordinated by a central party (*e.g.*, a ridesharing company) that has more information than any one of the players, but the players are free to ignore recommendations if they so choose.

*Persuasion in games* [17, 3, 23, 14, 30]. The mediator (in that literature, usually "sender") has more information than the players ("receivers"), and wishes to tell information to the receivers so as to persuade them to act in a certain way.

*Automated mechanism design* [6, 8, 33, 35, 26, 34, 18, 19]. Players have private information unknown to the mediator. The mediator wishes to commit to a strategy—that is, set a mechanism—such that players are incentivized to honestly reveal their information. In fact, in Appendix E we will see that we recover the polynomial-time Bayes-Nash randomized mechanism design algorithm of [6, 8] as a special case of our main result.

---

[1]We do not consider the NFCE, because it breaks our paradigm, which enforces that the mediator's recommendation be a single action. In NFCE, the whole strategy needs to be revealed upfront. It is an open question whether it is possible to even find *one* NFCE in polynomial time, not to mention an optimal one.

Some of the above examples are often used to motivate correlated equilibria. However, when the mediator is a rational agent with the ability to remember information that it is told and pass the information between players as necessary, we will argue that communication or certification equilibrium should be the notion of choice, for both conceptual and computational reasons.

## 2 Preliminaries

In this section, we discuss background on correlation in extensive-form games.

**Extensive-form games.** An *extensive-form game* $\Gamma$ with $n$ players consists of the following.

1. A directed tree of *nodes* or *histories* $\mathcal{H}$, whose root is denoted $\varnothing$. The depth of the tree will be denoted $T$. The edges out of nodes are labeled with *actions*, and the set of such actions will be denoted $A_h$. Given a node $h \in H$ and action $a$ at $h$, the child reached by following action $a$ at node $h$ is denoted $ha$. The set of terminal (leaf) nodes in $\mathcal{H}$ is denoted $\mathcal{Z}$. Terminal nodes will always be denoted $z$ throughout the paper.

2. A partition $\mathcal{H} \setminus \mathcal{Z} = \mathcal{H}_{\mathsf{C}} \sqcup \mathcal{H}_1 \sqcup \cdots \sqcup \ldots \mathcal{H}_n$ of nodes, where $\mathcal{H}_i$ is the set of all nodes at which player $i$ plays and player $\mathcal{H}_{\mathsf{C}}$ is the set of chance nodes.

3. For each player $i$, a partition $\mathcal{I}_i$ of player $i$'s decision nodes, $\mathcal{H}_i$, into *information sets* or *infosets*. Every node in a given information set $I$ must have the same set of actions, denoted $A_I$. We will call the partition $\mathcal{I} = \mathcal{I}_1 \sqcup \cdots \sqcup \mathcal{I}_n$ the *players' information partition*.

4. For each player $i$, a *utility vector* $\boldsymbol{u}_i \in [0,1]^{\mathcal{Z}}$, where $u_i[z]$ denotes the utility achieved by player $i$ at terminal node $z$.

5. For each chance node $h \in \mathcal{H}_{\mathsf{C}}$, a probability distribution $p(\cdot|h)$ over the children of $h$.

The *sequence* $\sigma_i(h)$ is the list of infosets reached by player $i$, and actions taken by the player $i$ at those infosets, on the $\varnothing \to h$ path, *not* including the infoset at $h$ itself (if any). We will assume that each player has *perfect recall*—that is, for each infoset $I$, the sequence of the player acting at $I$ should be the same for each node in $I$. We will denote this sequence $\sigma(I)$. In perfect-recall games, nonempty sequences will be identified by the last infoset-action pair $Ia$ in them.

We also will assume that games are *timeable* and *fixed-turn-order*, that is, information sets do not span multiple levels of the tree, and all nodes in the same layer of the tree belong to the same player[2].

We will use the following notation. The relation $\preceq$ denotes the natural precedence order induced by the tree $\mathcal{H}$: we write $h \preceq h'$ means that $h$ is an ancestor of $h'$ (or $h = h'$), and for sets $S, S'$, we say $S \preceq S'$ if there are some $h \in S, h' \in S'$ such that $h \preceq h'$. The binary operation $\wedge$ denotes the lowest common ancestor: $h \wedge h'$ is the lowest node $u$ such that $u \preceq h, h'$.

For sequences, $\boldsymbol{\sigma}(h) = (\sigma_1(h), \ldots, \sigma_n(h))$ denotes the *joint sequence* of all players at node $h$. $N(\sigma)$ denotes the set of possible *next* infosets following sequence $\sigma$, that is, $N(\sigma) = \{I : \sigma(I) = \sigma\}$. The set $\Sigma_i$ denotes the set of sequences of player $i$, and $\Sigma$ denotes the set of all sequences across all players (i.e., $\Sigma = \sqcup_i \Sigma_i$).

A *pure strategy* for a player $i$ is a selection of one action for each information set $I \in \mathcal{I}_i$. A *pure profile* is a tuple of pure strategies. A *correlated profile* is a distribution over pure profiles.

We will generally work with strategies in *realization form* (see e.g., Koller et al. [20]). Given a pure strategy $\boldsymbol{x}$, we say that $\boldsymbol{x}$ *plays to* $z \in \mathcal{Z}$ if $\boldsymbol{x}$ plays every action on the $\varnothing \to z$ path. We will call the vector $\boldsymbol{x} \in \{0,1\}^{\mathcal{Z}}$ the *realization form* of $\boldsymbol{x}$. The realization form of a mixed strategy is the appropriate convex combination. The set of mixed strategies forms a convex subset of $\mathbb{R}^{\mathcal{Z}}$ that, so long as the player has perfect recall, can be expressed using linearly many constraints and variables.

We will occasionally need to discuss changing information partitions of $\Gamma$. If $\mathcal{J} = \mathcal{J}_1 \sqcup \cdots \sqcup \mathcal{J}_n$ is another valid information partition, we will use $\Gamma^{\mathcal{J}}$ to denote the game $\Gamma$ with its information partition replaced by $\mathcal{J}$. We will also occasionally need to talk about multiple games simultaneously; where this is the case, we will mark attributes of the game the same as the game itself. For example, $\hat{\mathcal{H}}$ is the node set of game $\hat{\Gamma}$.

---

[2]Timeability is not without loss of generality, but any game for which the precedence order $\preceq$ is a partial order over infosets can be converted to a timeable game by adding dummy nodes. Given timeability, fixed-turn-order is without loss of generality, also by adding dummy nodes

**Communication and certification equilibria.** Here, we review definitions related to *communication equilibria*, following Forges [11] and later related papers.

**Definition 2.1.** Let $S$ be a space of possible *messages*. A *pure mediator strategy* is a map $d : S^{\leq T} \to S$, where $S^{\leq T}$ denotes the set of sequences in $S$ of length at most $T$. A *randomized mediator strategy* (hereafter simply *mediator strategy*) is a distribution over pure mediator strategies.

We will assume that the space of possible messages is large, but not exponentially so. In particular, we will assume that $\{\bot\} \cup \mathcal{I} \cup \bigcup_h A_h \subseteq S$ (*i.e.*, messages can at least be nothing, information, or actions)[3] and that $|S| \leq \text{poly}(|\mathcal{H}|)$. The latter assumption is mostly for cleanliness in stating results: we will give algorithms that need $S$ as an input that we wish to run in time $\text{poly}(|\mathcal{H}|)$.

A mediator strategy augments a game as follows. If the strategy is randomized, it first samples a pure strategy $d$, which is hidden from the players. At each timestep $t$, a player reaches a history $h$ at which she must act, and observes the infoset $I \ni h$. She sends a message $s_t \in S$ to the mediator. The mediator then sends a response $d(s_1, \ldots, s_t)$, which depends on the message $s_t$ as well as the messages sent by all other players prior to timestep $t$. Then, the player chooses her action $a \in A_h$. We will call the sequence of messages sent and received between the mediator and player $i$, the *transcript with player $i$*. A *communication equilibrium*[4] is a Nash equilibrium of the game $\Gamma$ augmented with a mediator strategy. The mediator is allowed to perform arbitrary communication with the players. In particular, the mediator is allowed to *pass information from one player to another*. Further, the players are free to send whatever messages they wish to the mediator, including false or empty messages. These two factors distinguish communication equilibria from notions of *correlated equilibria*. In Section 3.4 we will discuss this comparison in greater detail.

A useful property in the literature on communication equilibria is the *revelation principle* (*e.g.*, [11]). Informally, the revelation principle states that any outcome achievable by an *arbitrary* strategy profile can also be achieved by a *direct* strategy profile, in which the players tell the mediator all their information and are subsequently directly told by the mediator which action to play. In order to be a communication equilibrium, the players still must not have any incentive to deviate from the protocol. That is, the equilibrium must be *robust* to all messages that a player may attempt to send to the mediator, even if *in equilibrium* the player always sends the honest message.

Forges and Koessler [12] further introduced a form of equilibrium for Bayesian games which they called *certification equilibria*. In certification equilibria, the messages that a player may legally send are dependent on their information; as such, some messages that a player can send are *verifiable*. At each information set $I \in \mathcal{I}$, let $S_I \subseteq S$ denote the set of messages that the player at infoset $I$ may send to the mediator. We will always assume that $I \in S_I$ and $\bot \in S_I$ for all $I$. That is, all players always have the options of revealing their true information or revealing nothing.

## 3 Extensive-form $\mathcal{S}$-certification equilibria

The central notion of interest in this paper is a generalization of the notion of certification equilibria [12] to extensive-form games.

**Definition 3.1.** Given an extensive-form game $\Gamma$ and a family of valid message sets $\mathcal{S} = \{S_I : I \in \mathcal{I}\}$, an $\mathcal{S}$-*certification equilibrium* is a Nash equilibrium of the game augmented by a randomized mediator, in which each player at each information set $I$ is restricted to sending a message $s \in S_I$.

The existence of $\mathcal{S}$-certification equilibria follows from the existence of *Nash* equilibria, which are the special case where the mediator does nothing.

We will need one extra condition on the message sets, which is known as the *nested range condition* (NRC) [15]: if $I \in S_{I'}$, then $S_I \subseteq S_{I'}$. That is, if a player with information $I'$ can lie by pretending to have information $I$, then that player can also emulate any other message she would have been

---

[3]*A priori*, although the messages are given these names, they carry no semantic meaning. The revelation principle is used to assign natural meaning to the messages.

[4]Previous models of communication in games [11, 12] usually worked with a model in which players send messages, receive messages, and play moves *simultaneously*, rather than in sequence as in the extensive-game model that we use. The simultaneous-move model is easy to recreate in extensive form: by adding further "dummy nodes" at which players learn information but only have one legal action, we can effectively re-order when players ought to communicate their information to the mediator.

able to send at $I$. Equivalently, the honest message $I$ should be the *most certifiable* message that a player can send at infoset $I$. Our main result is the following.

**Theorem 3.2.** *Let $u_M \in \mathbb{R}^{\mathcal{Z}}$ be an arbitrary utility vector for the mediator. Then there is a polynomial-time algorithm that, given a game $\Gamma$ and a message set family $\mathcal{S}$ satisfying the nested range condition, computes an optimal $\mathcal{S}$-certification equilibrium, that is, one that maximizes $\mathbb{E}_z\, u_M[z]$ where the expectation is over playouts of the game under equilibrium.*

In particular, by setting $S_I = S$ for all $I$, Theorem 3.2 implies that optimal communication equilibria can be computed in polynomial time.

The rest of the paper is organized as follows. First, we will prove our main theorem. Along the way, we will demonstrate a form of revelation principle for $\mathcal{S}$-certification equilibria. We will then discuss comparisons to other known forms of equilibrium, including the extensive-form correlated equilibrium [29], and several other natural extensions of our model. Finally, we will show experimental results that compare the computational efficiency and social welfare of various notions of equilibrium on some experimental game instances.

## 3.1 Proof of Theorem 3.2: The single-deviator mediator-augmented game

In this section, we construct a game $\hat{\Gamma}$, with $n + 1$ players, that describes the game $\Gamma$ where the mediator has been added as an explicit player. This game has similar structure to the one used by Forges [11, Corollary 2], but, critically, has size polynomial in $|\mathcal{H}|$. This is due to two critical differences. First, the players are assumed to either send $\perp$, or send messages that mediator cannot immediately prove to be off-equilibrium. In particular, if the player's last message was $I$ and the mediator recommended action $a$ at $I$, the player must send a message $I'$ with $\sigma(I') = Ia$. If this is impossible, the player must send $\perp$. Therefore, in particular, we will assume that $S_I$ consists of only $\perp$ and information sets $I'$ at the same level as $I$. Second, only one player is allowed to deviate. Therefore, the strategy of the mediator is not defined in cases where two or more players deviate.

We now formalize $\hat{\Gamma}$. Nodes in $\hat{\Gamma}$ will be identified by tuples $(h, \boldsymbol{\tau}, r)$ where $h \in \mathcal{H}$ is a history in $\Gamma$, $\boldsymbol{\tau} = (\tau_1, \ldots, \tau_n)$ is the collection of transcripts with all players, and $r \in \{\text{REV}, \text{REC}, \text{ACT}\}$ is a *stage marker* that denotes whether the current state is one in which a player should be *revealing information* (REV), the mediator should be *recommending a move* (REC), or the player should be *selecting an action* (ACT). The progression of $\hat{\Gamma}$ is then defined as follows. We will use the notation $\boldsymbol{\tau}[i \cdot s]$ to denote appending message $s$ to $\tau_i$.

- The root node of $\hat{\Gamma}$ is $(\varnothing, (\varnothing, \ldots, \varnothing), \text{REV})$.

- Nodes $(z, \boldsymbol{\tau}, \text{REV})$ for $z \in \mathcal{Z}$ are also terminal in $\Gamma$. The mediator gets utility $u_M[z]$, where $u$ is the mediator's utility function as in Theorem 3.2. All other players $i$ get utility $u_i[z]$.

- Nodes $(h, \boldsymbol{\tau}, \text{REV})$ for non-terminal $h$ are decision nodes for the player $i$ who acts at $h$.
  1. If $i$ is chance, there is one valid transition, to $(h, \boldsymbol{\tau}, \text{ACT})$.
  2. If some other player $j \neq i$ has already deviated (i.e., $\sigma_j(h) \neq \tau_j$), there is one valid transition, to $(h, \boldsymbol{\tau}[i \cdot I], \text{REC})$ where $I \ni h$.
  3. If player $i$ has deviated or no one has deviated, then player $i$ observes the infoset $I \ni h$, and selects a legal message $I' \in S_I \cap (\{\perp\} \cup N(\tau_i))$ to send to the mediator[5]. Transition to $(h, \boldsymbol{\tau}[i \cdot I'], \text{REV})$.

- At $(h, \boldsymbol{\tau}, \text{REC})$ where $h \in \mathcal{H}_i$, the mediator observes the transcript $\tau_i$ and makes a *recommendation* $a$. If $\tau_i$ contains any $\perp$ messages, then $a = \perp$. Otherwise, $a$ is a legal action $a \in A_I$, where $I$ is the most recent message in $\tau_i$. Transition to $(h, \boldsymbol{\tau}[i \cdot a], \text{ACT})$.

- Nodes $(h, \boldsymbol{\tau}, \text{ACT})$ for non-terminal $h$ are decision nodes for the player $i$ who acts at $h$.
  1. If $i$ is chance, then chance samples a random action $a \sim p(\cdot|h)$. Transition to $(ha, \boldsymbol{\tau}, \text{REV})$.
  2. If some other player $j \neq i$ has already deviated, there is one valid transition, to $(ha, \boldsymbol{\tau}, \text{REC})$, where $a$ is the action sent by the mediator.

---

[5]If $\tau_i$ contains any $\perp$ messages, then we take $N(\tau_i) = \varnothing$

3. If player $i$ has deviated or no one has deviated, then player $i$ observes the transcript $\tau_i$, and selects an action $a' \in A_h$. Transition to $(ha', \boldsymbol{\tau}, \text{REV})$. The action $a'$ need not be the recommended action.

Since at most one player can ever deviate by construction, and the length of the transcripts are fixed because turn order is common knowledge, the transcripts $\boldsymbol{\tau}$ can be identified with *sequences* $\sigma_i$ of the deviated player, if any. We will make this identification: we will use the shorthand $h^{\sigma_i}$ to denote the history $(h, (\sigma_{-i}(h), \sigma_i), \text{REV})$, and $h^\perp$ for $(h, \boldsymbol{\sigma}(h), \text{REV})$ (i.e., no one has deviated yet). Therefore, in particular, this game has at most $O(|\mathcal{H}||\Sigma|)$ histories.

For each non-mediator player, there is a well-defined *direct strategy* $\hat{\boldsymbol{x}}_i^*$ for that player: always report her true information $I \ni h$, and always play the action recommended by the mediator. The goal of the mediator is to *find a strategy $\hat{\boldsymbol{x}}_\mathsf{M}$ for itself that maximizes its expected utility, subject to the constraint that each player's direct strategy is a best response*—that is, find $\hat{\boldsymbol{x}}_\mathsf{M}$ such that $(\hat{\boldsymbol{x}}_\mathsf{M}, \hat{\boldsymbol{x}}_1^*, \ldots, \hat{\boldsymbol{x}}_n^*)$ is a (strong) Stackelberg equilibrium of $\hat{\Gamma}$.

We claim that finding a mediator strategy $\hat{\boldsymbol{x}}_\mathsf{M}$ that is a strong Stackelberg equilibrium in $\hat{\Gamma}$ is equivalent to finding an optimal $\mathcal{S}$-certification equilibrium in $\Gamma$. We prove this in two parts. First, we prove a version of the revelation principle for $\mathcal{S}$-certification equilibria.

**Definition 3.3.** An $\mathcal{S}$-certification equilibrium is *direct* if it satisfies the following two properties.

1. (*Mediator directness*) If the transcript $\tau_i$ of a player $i$ is exactly some sequence of player $i$, and player $i$ sends an infoset $I$ with $\sigma(I) = \tau_i$, then the mediator replies with an action $a \in A_I$. Otherwise[6], the mediator replies $\perp$.

2. (*Player directness*) In equilibrium, players always send their true information $I$, and, upon receiving an action $a \in A_I$, always play that action.

**Proposition 3.4** (Revelation principle for $\mathcal{S}$-certification equilibria under NRC). *Assume that $\mathcal{S}$ satisfies the nested range condition. For any $\mathcal{S}$-certification equilibrium, there is a realization-equivalent direct equilibrium.*

Omitted proofs can be found in the appendix. Since direct mediator strategies are exactly the mediator strategies in $\hat{\Gamma}$, and the player strategies are only limited versions of what they are allowed to do in $\mathcal{S}$-certification equilibrium, this implies that, for any $\mathcal{S}$-certification equilibrium, there is a mediator strategy $\hat{\boldsymbol{x}}_\mathsf{M}$ in $\hat{\Gamma}$ such that $(\hat{\boldsymbol{x}}_\mathsf{M}, \hat{\boldsymbol{x}}_1^*, \ldots, \hat{\boldsymbol{x}}_n^*)$ is a Stackelberg equilibrium. We will also need the converse of this statement.

**Proposition 3.5.** *Let $\hat{\boldsymbol{x}}_\mathsf{M}$ be a strategy for the mediator in $\hat{\Gamma}$ such that, in the strategy profile $(\hat{\boldsymbol{x}}_\mathsf{M}, \hat{\boldsymbol{x}}_1^*, \ldots, \hat{\boldsymbol{x}}_n^*)$, every $\hat{\boldsymbol{x}}_i^*$ for $i \neq \mathsf{M}$ is a best response. Then there is a direct $\mathcal{S}$-certification equilibrium that is realization-equivalent to $(\hat{\boldsymbol{x}}_\mathsf{M}, \hat{\boldsymbol{x}}_1^*, \ldots, \hat{\boldsymbol{x}}_n^*)$.*

Therefore, we have shown that the mediator strategies $\hat{\boldsymbol{x}}_\mathsf{M}$ in $\hat{\Gamma}$ for which $(\hat{\boldsymbol{x}}_\mathsf{M}, \hat{\boldsymbol{x}}_1^*, \ldots, \hat{\boldsymbol{x}}_n^*)$ is a Stackelberg equilibrium in $\hat{\Gamma}$ correspond exactly to optimal $\mathcal{S}$-certification equilibria of $\Gamma$. Such a Stackelberg equilibrium can be found by solving the following program:

$$
\begin{aligned}
\max_{\hat{\boldsymbol{x}}_\mathsf{M} \in \hat{\mathcal{X}}_\mathsf{M}} \quad & \sum_{\hat{z} \in \hat{\mathcal{Z}}} \hat{x}_\mathsf{M}[\hat{z}] \hat{u}_\mathsf{M}[\hat{z}] \hat{p}(\hat{z}) \prod_{i \in [n]} \hat{x}_i^*[\hat{z}] \\
\text{s.t.} \quad & \max_{\hat{\boldsymbol{x}}_j' \in \hat{\mathcal{X}}_j} \sum_{\hat{z} \in \hat{\mathcal{Z}}} \hat{x}_\mathsf{M}[\hat{z}] \hat{u}_i[\hat{z}] \hat{p}(\hat{z}) \big( \hat{x}_j'[\hat{z}] - \hat{x}_j^*[\hat{z}] \big) \prod_{i \neq j} \hat{x}_i^*[\hat{z}] \leq 0 \quad \forall j \in [n]
\end{aligned}
\tag{1}
$$

where $\hat{\mathcal{X}}_i$ is the sequence-form strategy space [20] of player $i$ in $\hat{\Gamma}$.

The only variables in the program are $\hat{\boldsymbol{x}}_i$ for each player $i$ and the mediator. In particular, the direct strategies $\hat{\boldsymbol{x}}_i^*$ are constants. Therefore, the objective is a linear function, and the inner maximization constraints are bilinear in $\hat{\boldsymbol{x}}_\mathsf{M}$ and $\hat{\boldsymbol{x}}_j$. Therefore, this program can be converted to a linear program by dualizing the inner optimizations. For more details on this conversion, see Appendix B. The result is a linear program of size $O(n|\hat{\mathcal{H}}|) = O(n|\mathcal{H}||\Sigma|)$. We have thus proved Theorem 3.2.

---

[6]This condition is necessary because, if the mediator does not know what infoset the player is in, the mediator may not be *able* to send the player a valid action, because action sets may differ by infoset.

## 3.2 Extensions and special cases

In this section, we describe several extensions and interesting special cases of our main result.

**Full-certification equilibria.** One particular special case of $\mathcal{S}$-certification equilibria which is particularly useful. We define a *full-certification equilibrium* as an $\mathcal{S}$-certification equilibrium where $S_I = \{\bot, I\}$. Intuitively, this means that players cannot *lie* to the mediator, but they may *withhold* information. We will call such an equilibrium *full-certification*. Removing valid messages from the players only reduces their ability to deviate and thus increases the space of possible equilibrium strategies. As such, the full-certification equilibria are the largest class of $\mathcal{S}$-certification equilibria.

For full-certification equilibria, the size of game $\hat{\Gamma}$ reduces dramatically. Indeed, in all histories $h^{Ia}$ of $\hat{\Gamma}$, we must have $I \preceq h$. Therefore, we have $|\hat{\mathcal{H}}| \leq |\mathcal{H}|BD$ where $B$ is the maximum branching factor and $D$ is the depth of the game tree, *i.e.*, the size of $\hat{\Gamma}$ goes from essentially quadratic to essentially quasilinear in $|\mathcal{H}|$. The mediator's decision points in $\hat{\Gamma}$ for a full-certification equilibrium are the *trigger histories* used by Zhang et al. [31] in their analysis of various notions of correlated equilibria. Later, we will draw further connections between full certification and correlation.

**Changing the mediator's information.** In certain cases, the mediator, in addition to messages that it is sent by the players, also has its own observations about the world. These are trivial to incorporate into our model: simply change the information partition of the mediator in $\hat{\Gamma}$ as needed. Alternatively, one can imagine adding a "player", with no rewards (hence no incentive to deviate), whose sole purpose is to observe information and pass it to the mediator. For purposes of keeping the game small, it is easier to adopt the former method. To this end, consider any refinement partition $\mathcal{M}$ of the mediator infosets in $\hat{\Gamma}$, and consider the game $\hat{\Gamma}^{\mathcal{M}}$ created by replacing the mediator's information partition in $\hat{\Gamma}$ with $\mathcal{M}$. Then we make the following definition.

**Definition 3.6.** An $(\mathcal{S}, \mathcal{M})$-*certification equilibrium* of $\Gamma$ is a mediator strategy $\hat{x}_{\mathsf{M}}$ in $\hat{\Gamma}^{\mathcal{M}}$ such that, in the strategy profile $(\hat{x}_{\mathsf{M}}, \hat{x}_1^*, \ldots, \hat{x}_n^*)$, every $x_i^*$ for $i \neq \mathsf{M}$ is a best response.

$(\mathcal{S}, \mathcal{M})$-certification equilibria may not exist: indeed, if $\mathcal{M}$ is coarser than the mediator's original information partition in $\hat{\Gamma}$, then the mediator may not have enough information to provide good recommendations under the restrictions of $\hat{\Gamma}$. This can be remedied by allowing payments (see Appendix E), or by making the assumption that the mediator *at least* knows the transcript of the player to whom she is making any nontrivial recommendation:

**Definition 3.7.** A mediator partition $\mathcal{M}$ is *direct* if, at every mediator decision point $(h, \boldsymbol{\tau}, \mathrm{REC})$, so long as $|A_h| > 1$, the mediator knows the transcript of the player acting at $h$. $\mathcal{M}$ is *strongly direct* if the mediator also observes the transcript when $|A_h| = 1$.

The condition $|A_h| > 1$ in the definition allows the mediator to possibly *not* observe the full information of a player if she does not need to make a nontrivial recommendation to that player. In particular, this allows players to sometimes have information that they only partially reveal to the mediator, so long as the player does not immediately need to act on such information.

**Coarseness.** In literature on correlation, *coarseness* refers to the restriction that a player must obey any recommendation that she receives (but may choose to deviate by not requesting a recommendation and instead playing any other action). *Normal-form coarseness* further adds the restriction that players can only choose to deviate at the start of the game—the mediator essentially takes over and plays the game on behalf of non-deviating players. These notions can easily be expressed in terms of our augmented game, therefore also allowing us to express coarse versions of our equilibrium notions as augmented games.

## 3.3 The gap between polynomial and not polynomial

If players cannot send messages to the mediator at all, and the mediator has no other way of gaining any information, we recover the notion of *autonomous correlated equilibrium (ACE)*. It is NP-hard to compute optimal ACE, even in Bayesian games (see *e.g.*, von Stengel and Forges [29]).

When $\mathcal{M}$ is direct and perfect recall, computing an optimal *direct* $(\mathcal{S}, \mathcal{M})$-certification equilibrium can be done in polynomial time using our framework. When $\mathcal{S}$ obeys NRC and $\mathcal{M}$ satisfies a

stronger condition[7], the proof of the revelation principle (Propositions 3.4 and 3.5) works, and the resulting equilibrium is guaranteed to be optimal over all possible equilibria including those that may not be direct.

If NRC does not hold, one can still solve the program (1), and the solution is still guaranteed to be an optimal *direct* equilibrium by Proposition 3.5. However, it is not guaranteed to be optimal over all possible communication structures. Indeed, Green and Laffont [15, Theorem 1] give an instance in which, without NRC, there can be an outcome distribution that is not implementable by a direct mediator. Our program cannot find such an outcome distribution. The counterexample does not preclude the possibility of efficient algorithms for finding optimal certification equilibria in more general cases, but does give intuition for why NRC is crucial to our construction.

We could also consider changing the mediator's information partition so that the mediator does not have perfect recall. This transformation allows us to recover notions of *correlation* in games. Indeed, if we start from the *full-certification* equilibrium and only allow the mediator to remember the transcript with the player she is currently talking to, we recover EFCE. Adding coarseness similarly recovers EFCCE and NFCCE. In this setting, the inability to represent the strategy space of an imperfect-recall player may result in the loss of efficient algorithms.

### 3.4 A family of equilibria

By varying **1)** what the mediator observes, **2)** whether the mediator has perfect recall, **3)** whether the players can lie or only withhold information, and **4)** when and how players can deviate from the mediator's recommended actions, we can use our framework to define a family consisting of 16 conceptually different equilibrium notions. More can be generated by considering other variations in this design space, but we focus on the extreme cases in the table. Some of these were already defined in the literature; the remaining names are ours. The result is Table 1. An inclusion diagram for these notions can be found in Appendix G.

In the table, *ex ante* means that players have only a binary choice between deviating (in which case they can play whatever they want) and playing (in which case they must always be direct and obey recommendations). With *ex ante* deviations, it does not matter whether lying is allowed because we can never get to that stage: either the player deviates immediately and never communicates with the mediator, or the player is direct. If the mediator only remembers the current active player's information, and players cannot lie, withholding and coarsely deviating are the same.

*Mediator information advantage* means that the mediator always learns the infoset of the current active player, and therefore requires no messages from the players. This is equivalent to forcing players to truthfully report information. A mediator with information advantage may still not have perfect information—for example, it will not know whether a player (or nature) has played an action until some other player observes the action. In this setting, the mediator may also have extra private information (known to none of the players), leading to the setting of Bayesian persuasion [17]. In extensive-form games, there are two different reasonable notions of persuasion: one that stems from extending *correlated* equilibria, and one that stems from extending *communication* equilibria. The distinction is that, in the former, the mediator has imperfect recall. For a more in-depth discussion of Bayesian persuasion, see Appendix F.

Our framework allows optimal equilibria for all notions in the table to be computed. For perfect-recall mediators, this is possible in polynomial time via the sequence form; for imperfect-recall mediators, the problem is NP-hard, in general, but the *team belief DAG* of Zhang et al. [32] can be used to recover fixed-parameter algorithms. For the notions of correlated equilibrium, this method results in basically the same LP as the *correlation DAG* of Zhang et al. [31].

We do not claim that all of these notions are easy to motivate. For example, correlated equilibria are usually arrived at in the "truth known, imperfect recall" setting; the correlated equilibrium notions where lying is allowed are more difficult to motivate in this respect. Further, even the fixed-parameter algorithms of Zhang et al. [31] would fail in this setting, because "public states" can no longer be treated as public due to the possibility of lying players. We leave to future research the problem of finding a motivation for the notions that we do not reference elsewhere in the paper.

---

[7]Roughly speaking, this condition is that players should not be able to cause the mediator to gain information apart from their own messages by sending messages. It holds for all notions we discuss in this paper. Formalizing the general case is beyond the scope of this paper.

Table 1: A whole family of equilibria. See Section 3.4 for an explanation of the terms used in the table. NF, EF, and IR stand for normal-form, extensive-form, and imperfect-recall respectively.

| | | when can players deviate? | | |
|---|---|---|---|---|
| | | ex ante | ex interim | |
| | | | coarse | not coarse |
| **mediator remembers only current player's transcript** | lying possible | NFCCE [25] | truthful EFCCE | truthful EFCE |
| | withholding only | | EFCCE [10] | EFCE [29] |
| | mediator information advantage | NF coarse IR persuasion [3] | coarse IR persuasion | IR persuasion |
| **mediator perfect recall** | lying possible | NF coarse full-cert ("mediated" [24]) | coarse comm | comm [11] |
| | withholding only | | coarse full-cert | full-cert [12] |
| | mediator information advantage | NF coarse persuasion | coarse persuasion | persuasion |

# 4 Experiments

We ran our algorithm for communication and full-certification equilibria on various two-player games, and compared the results to those given by notions of optimal correlation in games. The games used in the experiments are given in Appendix D. All experiments were allocated four CPU cores and 64 GB of RAM. Linear programs were solved with Gurobi 9.5. When payments are used, the allowable payment range is $[0, M]$ where $M$ is the reward range of the game. Experimental results can be found in Table 2.

In the *battleship* and *sheriff* instances, there is not a significant difference in performance between finding full-certification equilibria and finding optimal correlated equilibria in terms of performance—this is because, unlike in the general case, optimal correlated equilibria in two-player games without chance can be found in polynomial time [29] anyway. In the *ridesharing* instances,

Table 2: Table of experimental results. Values are the optimal social welfare given the type of equilibrium. Values and timings for optimal correlated equilibria were taken from Zhang et al. [31] and are included here for purposes of comparison. When payments are used, the mediator is informed before making the payment of whether the player was honest, and the optimization objective is the social welfare of the original terminal state, minus any payments made. "oom" is out of memory.

| game | $|\mathcal{Z}|$ | | [ZFCS'22] NFCCE | EFCCE | EFCE | *This paper* NF Coarse Cert no pay | pay | Coarse Cert no pay | pay | Cert no pay | pay | Comm no pay | pay |
|---|---|---|---|---|---|---|---|---|---|---|---|---|---|
| B222 | 1072 | value | 0.000 | -0.525 | -0.525 | 0.000 | 0.000 | -0.525 | -0.333 | -0.525 | -0.453 | -0.750 | -0.520 |
| | | time | 0.02s | 0.05s | 0.17s | 0.01s | 0.01s | 0.02s | 0.07s | 0.06s | 0.17s | 3.80s | 4.05s |
| B322 | 19116 | value | 0.000 | -0.317 | -0.317 | -0.000 | 0.000 | -0.317 | -0.200 | -0.317 | -0.226 | oom | oom |
| | | time | 0.21s | 1.38s | 5.83s | 0.02s | 0.15s | 0.05s | 0.72s | 0.28s | 4.05s | oom | oom |
| B323 | 191916 | value | 0.000 | -0.375 | -0.375 | 0.000 | 0.000 | -0.375 | -0.250 | -0.375 | oom | oom | oom |
| | | time | 2.82s | 32.84s | 1m 55s | 0.32s | 2.42s | 2.77s | 16.50s | 22.59s | oom | oom | oom |
| S122 | 396 | value | 13.636 | 9.565 | 9.078 | 50.000 | 50.000 | 10.000 | 42.000 | 10.000 | 42.000 | 0.820 | 42.000 |
| | | time | 0.01s | 0.02s | 0.04s | 0.01s | 0.01s | 0.02s | 0.11s | 0.08s | 0.20s | 0.85s | 1.74s |
| S123 | 2376 | value | 13.636 | 10.000 | 10.000 | 50.000 | 50.000 | 10.000 | 42.000 | 10.000 | 42.000 | 0.820 | 42.000 |
| | | time | 0.04s | 0.23s | 0.65s | 0.03s | 0.07s | 0.12s | 0.25s | 0.46s | 1.04s | 1m 13s | 1m 49s |
| S133 | 5632 | value | 18.182 | 15.000 | 15.000 | 50.000 | 50.000 | 15.000 | 43.000 | 15.000 | 43.000 | 0.820 | oom |
| | | time | 0.04s | 1.51s | 2.46s | 0.06s | 0.12s | 0.31s | 0.65s | 1.78s | 2.89s | 17m 7s | oom |
| RS12 | 400 | value | 6.010 | 6.010 | 6.010 | 6.173 | 6.173 | 6.173 | 6.173 | 6.173 | 6.173 | 6.173 | 6.173 |
| | | time | 0.02s | 0.01s | 0.01s | 0.00s | 0.01s | 0.00s | 0.01s | 0.01s | 0.04s | 0.91s | 1.77s |
| RS13 | 4356 | value | 9.398 | 9.385 | 9.367 | 9.622 | 9.622 | 9.622 | 9.622 | 9.622 | 9.622 | 9.592 | 9.592 |
| | | time | 2.82s | 1m 28s | 12m 31s | 0.03s | 0.11s | 0.07s | 0.19s | 0.16s | 0.55s | 3m 9s | 6m 42s |
| RS14 | 229888 | value | oom | oom | oom | 10.500 | 10.500 | 10.500 | 10.500 | 10.500 | 10.500 | oom | oom |
| | | time | oom | oom | oom | 0.20s | 1.41s | 0.66s | 3.97s | 2.34s | 12.04s | oom | oom |
| RS22 | 484 | value | 7.188 | 7.176 | 7.176 | 7.594 | 7.594 | 7.594 | 7.594 | 7.594 | 7.594 | 7.594 | 7.594 |
| | | time | 0.20s | 0.20s | 0.16s | 0.00s | 0.01s | 0.01s | 0.02s | 0.01s | 0.04s | 0.90s | 1.80s |
| RS23 | 4096 | value | 10.961 | 10.820 | 10.791 | 11.516 | 11.516 | 11.513 | 11.513 | 11.485 | 11.485 | 11.464 | 11.464 |
| | | time | 3.12s | 56m 31s | 6m 35s | 0.03s | 0.10s | 0.06s | 0.18s | 0.19s | 0.63s | 7m 43s | 12m 1s |

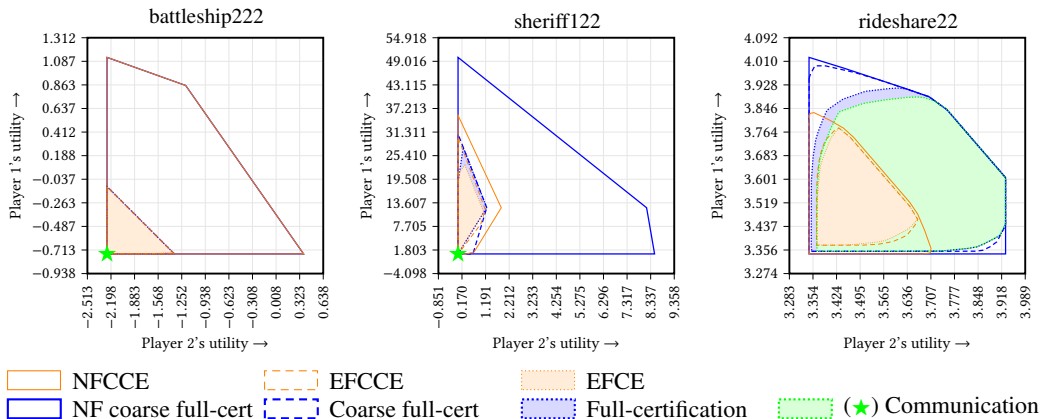

*Figure 1: Payoff spaces for various games and notions of equilibrium. The symbol ★ indicates that the set of communication equilibrium payoffs for that game is (at least, modulo numerical precision) that single point. In the battleship instance, many of the notions overlap.*

computing optimal correlated equilibria is much more computationally intensive because the game contains non-public chance actions. Computing optimal full-certification equilibria is comparably easy, and this difference is clearly seen in the timing results.

Finding optimal *communication* equilibria is much more intensive than finding optimal full-certification equilibria, owing to the quadratic size of the augmented game for communication equilibria. This often causes communication equilibria to be the *hardest* of the notions to compute in practice, despite optimal correlation being NP-hard.

In Figure 1, we have plotted the payoff spaces of some representative instances. The plots show how the polytopes of communication and full-certification equilibria behave relative to correlated equilibria. In the *battleship* and *sheriff* instances, the space of communication equilibrium payoffs is a single point, which implies that the space of NFCE (and hence Nash) equilibrium payoffs is also that single point. Unfortunately, that point is the Pareto-least-optimal point in the space of EFCEs. In the *ridesharing* instances, communication allows higher payoffs. This is because the mediator is allowed to "leak" information between players.

## 5 Conclusions and future research

We have shown that optimal communication and certification equilibria in extensive-form games can be computed via linear programs of polynomial size, or almost-linear size in the full-certification case. We have used our machinery to derive an entire family of equilibrium concepts which we hope to be of use in the future.

Possible future directions include the following.

1. Are there efficient *online learning dynamics*, in any reasonable sense of that term, that converge to certification or communication equilibrium?

2. Is there a better-than-quadratic-size linear program for communication equilibria?

3. Is it possible to extend our augmented game construction to also cover *normal-form* correlated equilibria while maintaining efficiency?

4. Investigate further the comparison between communication and correlation in games. For example, when and why do communication equilibria achieve higher social welfare than extensive-form correlated equilibria?

## Acknowledgements

This material is based on work supported by the National Science Foundation under grants IIS-1901403 and CCF-1733556, and the ARO under award W911NF2010081.

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
