# A  Omitted proofs

## A.1  Proposition 3.4

*Proof.* We follow the usual structure of revelation principle proofs. Given an $\mathcal{S}$-certification equilibrium, consider augmenting the mediator $d$ by devices $d^i$, one per player, that acts as follows. $d^i$ internally keeps track of the current sequence $\sigma_i$ of player $i$. Whenever it is player $i$'s turn to act, $d^i$ expects to be told an information set $I'$ with $\sigma(I') = Ia$. Then $d^i$ samples what message $s$ player $i$ would have honestly sent to the mediator at infoset $I'$, and forwards this to $d$. When $d$ replies with a message $s'$, $d^i$ samples the action $a'$ the player would have played, sends that to the player, and updates her internal state to $\sigma_i := I'a'$. If $d^i$ ever receives an invalid message (i.e., anything except an infoset $I'$ with $\sigma(I') = Ia$), it resorts to always sending $\perp$ to both $d$ and player $i$ for the remainder of the game.

To see that this results in an equilibrium, note that player $i$ can always simulate $d^i$ when playing with the original mediator $d$—therefore, any deviation she can perform against the direct equilibrium can also be performed with $d$ (This is where NRC is used: since the acting player may in reality have lied about her information, without the NRC, it could be the case that the simulated message $s$ would not be legal for player $i$ to send). $\qquad\square$

## A.2  Proposition 3.5

*Proof.* Consider a mediator for the full game that acts according to $\hat{x}_{\mathsf{M}}$. If the player sends messages that are not in $\hat{\Gamma}$, then the mediator acts as if the player sent $\perp$. If the mediator reaches a state in which at least two players have provably deviated (i.e., the mediator has reached a state not in $\hat{\Gamma}$), then the mediator acts arbitrarily. This mediator strategy, along with the players' direct strategies, forms a strategy profile. The only difference between $\hat{\Gamma}$ and the true communication protocol induced by $\mathcal{S}$ is, in $\hat{\Gamma}$, players are not able to send certain messages. But, in any case, those messages are ones that are never sent by an honest player; therefore, the mediator will always act as if the player sent $\perp$ if it receives such a message. Therefore, the player gains nothing by having extra messages that she can send. This completes the proof. $\qquad\square$

# B  Full linear program

In this section, we present the full LP formulation implied by the discussion in Section 3.1.

Dropping the hats for notational cleanliness, and letting $\mathcal{X}_j = \{\boldsymbol{x}_j : \boldsymbol{F}_j \boldsymbol{x}_j = \boldsymbol{f}_j, \boldsymbol{x}_j \geq 0\}$ be the realization-form representation of player $j$'s decision space, the program (1) has the form

$$\max_{\boldsymbol{x}_{\mathsf{M}} \in \mathcal{X}_{\mathsf{M}}} \boldsymbol{c}^\top \boldsymbol{x} \quad \text{s.t.} \quad \max_{\boldsymbol{x}_j : \boldsymbol{F}_j \boldsymbol{x}_j = \boldsymbol{f}_j, \boldsymbol{x}_j \geq 0} \boldsymbol{x}_{\mathsf{M}}^\top \boldsymbol{A}_j \boldsymbol{x}_j \leq 0. \ \ \forall j \in [n]$$

Dualizing the inner maximizations, we have the linear program

$$\max \ \boldsymbol{c}^\top \boldsymbol{x}_{\mathsf{M}}$$
$$\text{s.t.} \quad \boldsymbol{F}_j^\top \boldsymbol{v}_j \geq \boldsymbol{A}_j^\top \boldsymbol{x}_{\mathsf{M}}, \ \ \boldsymbol{f}_j^\top \boldsymbol{v}_j \leq 0 \ \ \forall j \in [n]$$
$$\boldsymbol{x}_{\mathsf{M}} \in \mathcal{X}_{\mathsf{M}}$$

where $\boldsymbol{v}_j$ are dual variables. Intuitively, the dual variables represent *best-response values*: the vector $\boldsymbol{v}_j$ will be indexed by information sets $I$ for player $j$, and $v_j[I]$ will be the best-response value for player $j$ at infoset $I$ (assuming all other players are direct). This sort of dualization of an inner optimization problem to create a linear program is fairly standard—see Koller et al. [20] for an application to two-player zero-sum games and Farina et al. [9] for the same approach applied to the special case of correlation.

# C  Example of game in which full-certification equilibria dominate NFCCEs

We now give an example that demonstrates the difference between NFCCEs (where the mediator has imperfect recall) and the perfect-recall notions. In particular, we will show a game where there is an

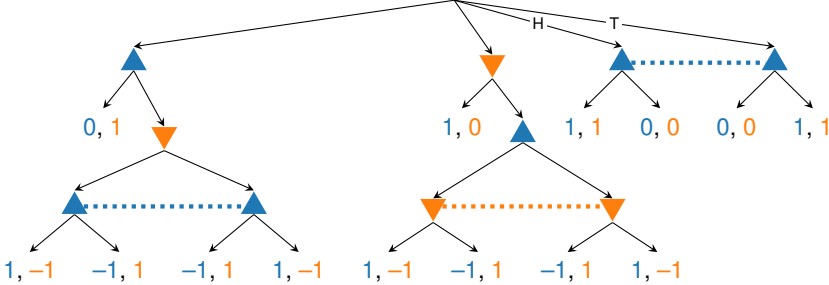

*Figure 2: The counterexample in Appendix C. At terminal nodes, ▲'s utility is listed first. The root node is a chance node, and the distribution at it is uniform random.*

(extensive-form) persuasion strategy for the mediator that leads to outcomes Pareto-dominating every NFCCE of $\Gamma$. Since persuasion can be recovered from full certification (or even communication) by adding a dummy player who communicates information to the mediator and has no incentives (thus is always direct), this also shows that communication equilibria can Pareto-dominate NFCCEs.

Consider the game whose tree is depicted in Figure 2. At the root, chance selects a subgame $s \in \{▲, ▼, \mathsf{H}, \mathsf{T}\}$. If chance selects $s \in \{▲, ▼\}$, then player $s$ is offered a chance to exit the game; if she does, she scores 0 and the other player scores 1. If she does not exit, the two players play matching pennies with the *other* player (not $s$) playing first (this will be relevant when we take perfect-recall refinements). The winner scores 1 point and the loser $-1$. If chance selects $s \in \{\mathsf{H}, \mathsf{T}\}$, then ▲ attempts to guess which one was selected. If successful, both players score 1 point; otherwise, both scores 0. The following properties can be directly verified:

- For every NFCCE in both $\Gamma$ and its perfect-information refinement, both ▲ and ▼ have value exactly $1/2$.

- The following mediator pure strategy is persuasive in $\Gamma$, where both players achieve value $3/4$: both players exit when possible, and if $s \in \{\mathsf{H}, \mathsf{T}\}$, the mediator informs ▲ which is the case. If the matching pennies subgame is entered, the mediator recommends that each player play independently and uniformly at random.

Intuitively, this outcome exists because the mediator has the power to selectively tell players information when it is to their benefit (*e.g.* in the case $s \in \{\mathsf{H}, \mathsf{T}\}$, where the mediator can inform ▲ of chance's choice to allow ▲ to score a point) and withhold information to incentivize cooperation (*e.g.*, in the case $s = ▲$, where the mediator can withhold ▼'s matching pennies bit from ▲ to prevent ▲ from deviating).

## D  Descriptions of games in experiments

- **B** is a two-player Battleship game [9] with a grid of size $h \times w$ and $r$ rounds, made into a nonzero-sum game by having each player value their own ships greater than their opponents'. The three numbers afterward are, in order, the length, width, and number of shots per side. Each player has a single ship of unit size.

- **S** is a simplified Sheriff of Nottingham game [9], a small game modeling a negotiation between two players. The first two numbers roughly correspond to how much power each player has in negotiation; the final number is the number of rounds of negotiation.

- **RS** is a small ridesharing game [31], played on a graph, in which the players walk around the graph attempting to serve "customers" who reside at certain nodes. The first number (1 or 2) denotes the specific graph on which the game is played; the second number is the number of steps in the game.

# E    Mechanisms with payments

The mediator may be able to take its own actions during the game, separate from the players. In our framework, this can be modelled by introducing an auxiliary player that has no incentives of her own and therefore no reason to disobey the mediator recommendations. One particular use case of such a player is *payments*: the auxiliary player can collect payments from or give payments to players, as a means of incentivizing players to perform certain actions.

We can make this concrete as follows. At the end of the game $\hat{\Gamma}$, chance picks a random player $i^* \in [n]$, reveals it to the mediator. The mediator then selects a payment $p \in \{L, U\}$ to be given to the player, where $L, U \in \mathbb{R}$ are a minimum or maximum payment (that may be functions of the mediator's information). The player gains utility $np$ and the mediator loses utility $np$. By varying the probability with which the mediator makes the two payments, the mediator can (in expectation) pay any player any amount in the range $[L, U]$. The size of the mediator-augmented game increases by a factor of $O(n)$.

If $U - L$ is at least the reward range of the game, and the mediator always learns whether players have acted directly, then any strategy at all can be enforced in equilibrium simply by giving each player a sufficiently large payoff for cooperating. The ability of the mediator to commit to a strategy is critical when payments are involved: if a mediator could not credibly commit, then it would never make a payment because it has negative incentive to do so.

**Automated mechanism design.** A consequence of the above analysis is that we recover the Bayes-Nash[8] randomized automated mechanism design algorithm of Conitzer and Sandholm [6, 8] as a special case, as follows. That paper considers a Bayesian game with $T$ types per player and $n$ players. A type assignment $\boldsymbol{\theta} \in [T]^n$ is sampled from some joint distribution, and each type $\theta_i$ is revealed privately to player $i$ (for *ex interim* incentive compatibility) or publicly to all players (for *ex post* incentive compatibility). A round of communication ensues, in which each player informs the mediator about her own type (or lies about it). The mediator then chooses an outcome $o \in [O]$, and each player receives a utility that is a function of their own type $\theta_i$ and the outcome $o$. The resulting game has size $T^n O$, and each player has at most $T^n$ sequences, so the overall LP (1), after including payments, has size $\text{poly}(O, T^n)$, which is the same result observed by Conitzer and Sandholm [8].

**Costly messaging.** Suppose that players have a strict preference for not revealing information over revealing it, or for sending certain messages over others. In the game $\hat{\Gamma}$, we can easily express this preference by changing the utilities of player $i$. In this setting, direct equilibria may not exist in the general case: indeed, consider a single-player game where the player has perfect information and therefore no need to tell the mediator anything. Nonetheless, direct equilibria can still exist in some games if the mediator has some power to persuade the players to reveal their information—that is, if a player's information will help the mediator to give the player a better outcome. If the mediator knows whether players have been direct, there always exist payments large enough to incentivize direct behavior (or, indeed, *any* behavior), so payments can also be used to recover equilibrium existence.

This algorithm does not take into account the possibility that the mediator or players may be better off *not knowing* certain information than paying the price of gathering it, which is often the case in preference elicitation [*e.g.*, 21, 27] and has more recently been studied for automated mechanism design Zhang et al. [34], Kephart and Conitzer [18, 19]; it will only compute the optimal equilibrium for which the mediator gives sufficient incentive for all players to always honestly reveal information. Finding optimal equilibria in settings when the revelation principle fails is beyond the scope of this paper.

# F    Additional related research

Our results are extremely general across settings and applications, most notably mechanism design and Bayesian persuasion (information design). Many special cases of our main algorithm have been discussed in the literature. Here, we give an overview of some of them.

---

[8]There is no correlation to speak of, because players do not have any actions—they merely report information. Hence, we are able to discuss Bayes-Nash instead of Bayes-correlated equilibria.

**Automated multi-stage mechanism design.** In mechanism design, an uninformed mediator (the mechanism) takes actions based on information given to them by player(s) who have information but cannot take actions. In the previous section, we showed how our framework can be used to generalize the automated mechanism design algorithm of Conitzer and Sandholm [6]. We will now discuss several other very recent papers on automated mechanism design in dynamic settings, and how our paper relates to them.

Zhang and Conitzer [33] consider an automated dynamic mechanism design setting in a Markov game with one player. Their main positive result is an LP for the setting of short-horizon MDPs, which can be viewed as a special case of our framework by simply unrolling the (short-horizon) MDP into an extensive-form game.

Zhang et al. [35] consider an automated mechanism design setting in which the agent's only power is, at each timestep, to *quit* the decision process. They study Markov games, which in this setting are significantly involved difficult than extensive-form games. In extensive form, their setting can be formulated as an augmented game very similar to our framework, in which the "exit" action is explicitly added into the augmented game and the principal selects the action.

Papadimitriou et al. [26] study a dynamic auction design setting. In their most general setup, there are $k$ players (bidders) with independent valuations for each of $D$ items to be sold in sequence. The agents in their setting know *all* their valuations upfront, but only *report* their valuations for the item currently being sold. That constraint can be expressed in our framework in the language of $(\mathcal{S}, \mathcal{M})$-certification equilibria, by setting $\mathcal{M}$ such that the mediator only learns the player's valuation of the current item. As such, that setting is also a special case of ours—in particular, our algorithm matches their positive results, Theorems 7 and 8, that use a "dynamic programming LP" to compute the optimal randomized mechanism.

Sandholm et al. [28] study multi-stage mechanism design, but they use multiple stages for the purpose of reducing the amount of communication necessary for preference elicitation in what would otherwise be a single-stage setting. Their work is therefore orthogonal to ours.

**Automated mechanism design with partially-verifiable types.** Zhang et al. [34] give an algorithm for finding the optimal *direct* mechanism in a single-agent, single-stage setting with partially-verifiable types. The positive results in their paper focus on the case when all types have the same preferences over outcomes. Our work differs from that one in that we consider extensive-form (multi-stage) settings and a more general setup (in which the players can also take actions and can have arbitrary utility functions). However, we share in common with that paper the fact that we only compute the optimal *direct* mechanism, and hence rely on the revelation principle holding for that mechanism to be optimal across communication structures.

Kephart and Conitzer [18] analyze mechanism design in a single-stage, single-agent setting with costly reporting and not assuming the revelation principle. Their goal is to, given a social choice function, determine whether that function can be implemented. They analyze a large spectrum of cases, and discuss for each case whether this implementation problem is easy or (NP-)hard. In a follow-up paper [19], the same authors carefully investigate when the revelation principle does or does not hold (still in the single-stage, single-agent setting with reporting costs). When the revelation principle does hold, as we have discussed in Appendix E, our framework matches, as a special case, the polynomial-time algorithm of Conitzer and Sandholm [6, 8], and both can be used to compute whether a social choice function is implementable, simply by adding the appropriate linear constraints to the linear program formulation. However, when the revelation principle fails, that approach will fail to find an implementation for any social choice function that cannot be implemented by a direct mechanism.

**Automated multi-stage Bayesian persuasion (information design).** In Bayesian persuasion, also commonly referred to as information resign [17], the roles of the mediator and player are reversed compared to automated mechanism design: the mediator ("principal") has informational advantage, and the player(s) take the actions. The difference between the two settings lies in who has the commitment power: in automated mechanism design, the side with the power to take actions has the commitment power.

Celli et al. [3] study persuasion in extensive-form games with multiple players, in which the principal can only send a single signal at the beginning of the game to the players. They focus on a setting where the mediator can only send a single signal to the players, which is far more restrictive than our

setup of persuasion. Their setting of Bayes *coarse*-correlated equilibria is equivalent to what we call *normal-form coarse imperfect-recall persuasion*, and their results in this setting fall out as special case of our framework (using the correlation DAG of Zhang et al. [31] to represent the mediator's decision space). Their discussion of Bayes-*correlated* equilibria is not captured by our framework for the same reason that NFCEs are not captured.

Gan et al. [14] study persuasion in Markov games with a single player, in which the principal can send messages to the player at every time step. The extensive-form analogue of their setting is, once again, a special case of our framework. The complexity gap between myopic, advice-myopic, and far-sighted players, discussed in that paper for Markov games, disappears in extensive-form games because extensive form naturally allows history-dependent strategies for both the mediator and the players.

Wu et al. [30] study persuasion in Markov games with myopic players. That is, a new player arrives at every timestep. After performing a single action, the player gains a reward dependent on the true state (which, of course, the player may not actually know) and her action, and then leaves the system forever. The authors devise an online learning algorithm that provably has low regret for the sender. Compared to their setting, our setting is significantly more general—allowing multiple non-myopic agents as well as other forms of limited information and communication—but ours is based on linear programming instead of online learning, and works with extensive-form games instead of Markov games.

We are not the first to point out associations between communication equilibria and other settings such as mechanism design and persuasion. In particular, Bergemann and Morris [2] discuss in great depth the relationship between communication equilibria, mechanism design, and persuasion in Bayesian games. On top of that paper, our contribution is that we extend their ideas of unification to the more general setting of arbitrary extensive-form games and provide efficient algorithms for all of those cases.

**Preference elicitation from multiple agents.** There has been significant work on preference elicitation from multiple agents, starting in the context of combinatorial auctions [5]. Often the elicitor knows that some preference information from an agent is not needed for optimally allocating (and pricing) given what other agents have already revealed about their preferences. This is the case already in private-values settings. In quasilinear settings, if the elicitor receives enough information from the agents to determine the welfare-maximizing allocation and the Vickrey-Clarke-Groves payments, it is an *ex post* equilibrium for all players to tell the truth [5]. This is despite the fact that the elicitor's queries leak information across agents. Similar leakage also happens in preference elicitation in voting, and there are certain restrictions that can be imposed on the elicitation policy that eliminate any strategic effects that stem from that [7].

Our paradigm is fundamentally different from the above preference elicitation setting, in several ways. First, all our complexity results are at least linear in the size of the game tree, whereas in a combinatorial auction, the auctioneer (mediator) may pick one of exponentially many outcomes. Second, we always allow mixed strategies, which are typically not used in preference elicitation. Finally, in voting settings one often desires an equilibrium in *weakly dominant* strategies, which our paradigm also does not cover.

# G  Inclusion hierarchy

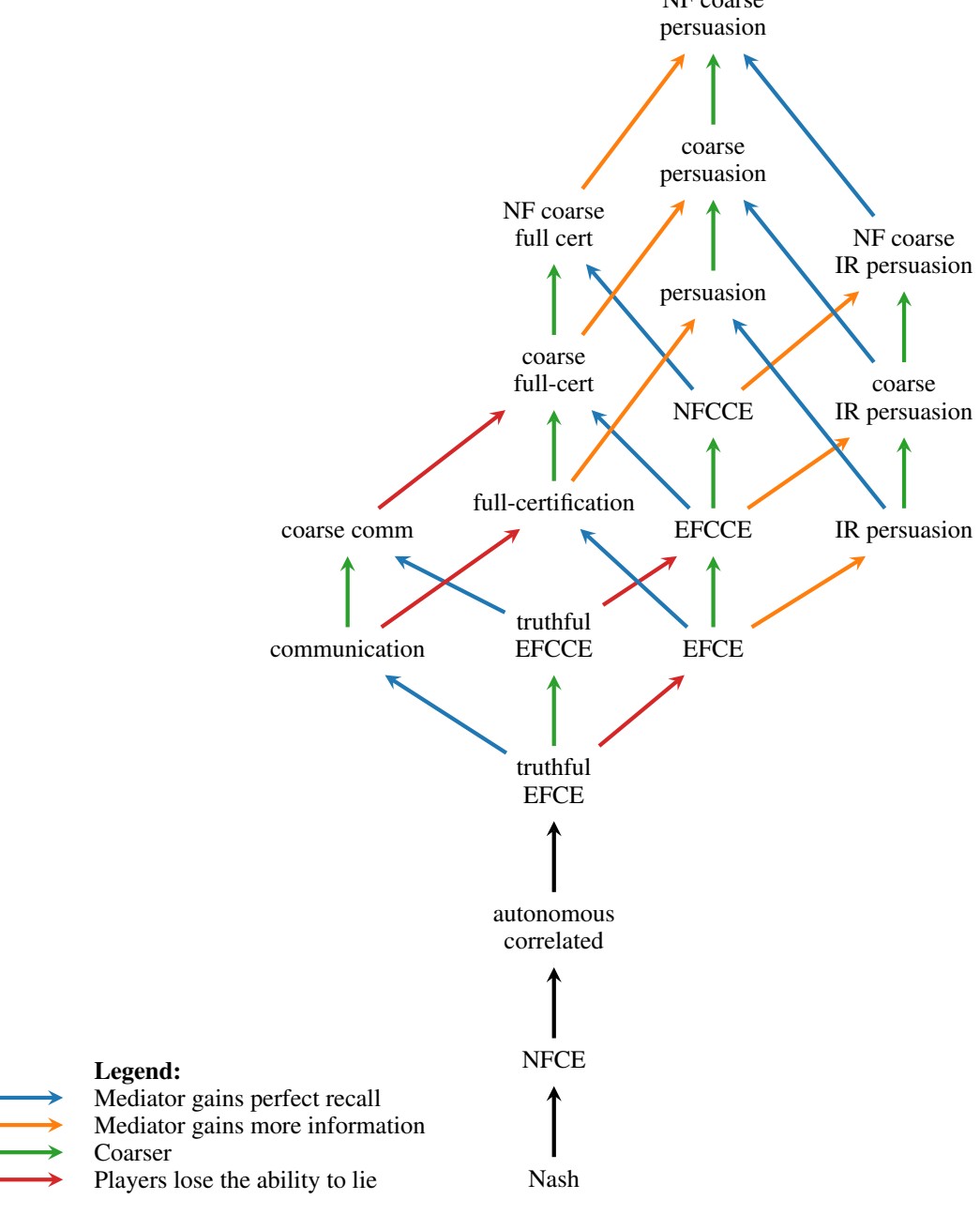

Figure 3: Inclusion diagram for the equilibrium notions in Section 3.4 and a few others. Autonomous correlated equilibria [11, 29] are equilibria in which the mediator cannot receive information from the players but still only gives recommendations one timestep at a time.