# OpenReview forum: "Polynomial-Time Optimal Equilibria with a Mediator in Extensive-Form Games"
_NeurIPS.cc/2022/Conference — NeurIPS 2022 Accept_

### Official Review · Reviewer_GECr · 2022-07-10

**Rating:** 7
**Confidence:** 3
**Soundness:** 3 good
**Presentation:** 3 good
**Contribution:** 3 good

**Summary:**

This paper studies the problem of computing optimal equilibria in extensive-form games. Standard equilibrium notions are hard (i.e., NP-hard) to optimize (e.g., for revenue or social welfare) in extensive-form games. The authors study communication and certification equilibria, showing that they can be optimized in polynomial time. These arise when a mediator is present who can send and receive messages from the players. They build additional results on top of this main result, namely:
- A class of equilibria where the players cannot reveal false information, leading to a linear size linear program (scales better than other notions).
- They consider the integration of payments, allowing the mediator a more expressive way to modify the outcome in settings where they are permitted.
- They define a family of equilibria from the main result that correspond to different tweaks to the setting, e.g., imperfect recall for moderator. They recover existing algorithms for equilibrium computation as special cases of their general method.

**Questions:**

N/A

**Limitations:**

Negative social impact is not a concern here, in my opinion. There is little explicit discussion of limitatins.

**Strengths And Weaknesses:**

Originality:
- They are the first to show that communication and certification equilibria can be optimized in polynomial time for extensive-form games.
- The paper lays out a map of equilibrium concepts and shows that many can be computed through the same meta-algorithm.
- Coverage of related work seems adequate.

Quality:
- There is a substantial amount of math I did not check, but the work appears technically sound.
- It's a theory paper, so not a lot of attention is spent on thinking about potentially real systems. I found myself wondering at some points wishing for a more direct discussion of motivation.
- There is not a lot of discussion of weaknesses.
- The authors' claims seem well supported.

Clarity:
- It's math-heavy. The notation was explained pretty clearly and a substantial chunk of the paper is spent doing so. I would prefer more pictures or examples to help explain (perhaps in the appendix due to space constraints).
- The authors say they will not release code, which I don't understand.

Significance:
- The results seem like they could be useful in some real-world contexts. In any case, it is good to know that such tractable families of equilibria exist.
- The generalization of past algorithms provides a bit of interesting big picture structure.

---

> ### Author Response · Authors · 2022-08-02
> **Response to review**
>
> Thanks for the review!
>
> **I would prefer more pictures or examples to help explain (perhaps in the appendix due to space constraints).**
>
> Other reviewers have also asked for more intuition/explanation in various places, and we have provided some in our responses to those reviewers. We will also include more examples and intuition in the final version of the paper using the extra page. In the paper, there is an example in Appendix B that is used to show that certification equilibria can dominate NFCCEs.

---

### Official Review · Reviewer_Xqpy · 2022-07-10

**Rating:** 5
**Confidence:** 5
**Soundness:** 2 fair
**Presentation:** 3 good
**Contribution:** 2 fair

**Summary:**

This paper studies extensive-form games augmented with a mediator that receives and sends messages from and to the agents involved in the game. In the particular case in which only the mediator can send messages and these can be interpreted as action recommendations, such an extended game resembles the one in the definition of correlated equilibria. At the same time, allowing for messages from the agents to the mediator allows to capture different notions of equilibrium, such as the communication equilibrium and the certification equilibrium. By leveraging such an augmented representation, the paper provides a linear programming formulation for finding optimal communication/certification equilibria in n-player extensive-form games in polynomial time. The theoretical results are complemented with an extensive experimental evaluation on a standard testbed of games.


**Questions:**

1) Can you provide more details as to why you can restrict the space of messages by using the revelation principle (see footnote 3)? I think more details on this are need in the paper.

2) It is not clear to me what you mean by “truthful EFCE” and other “truthful” concepts. Can you provide some more details on this.

**Limitations:**

Yes.

**Strengths And Weaknesses:**

ORIGINALITY

(+) The paper studies equilibrium concepts (communication and certification equilibria) that have never been addressed before by the algorithmic game theory community.

(-) The techniques employed by the authors are fairly standard, since they consist in a simple linear programming formulation.

QUALITY

(+) As far as I am concerned the results are sound.

(-) I think that more formalism in proving the technical results is needed. There are only two propositions in the paper, whose proofs given in the appendix are not sufficiently formalized in mathematical terms. Moreover, I think a proof is also need in order to shows the correctness of the linear programming formulation.

CLARITY

(+) Overall, the paper is well written and easy to follow.

(-) There are only some minor problems that I list in the following:

	(*) Line 28: Cite [24] after “extensive-form”.

	(*) In paragraph “Applications and related work.”: You should cite the recent works on learning dynamics converging to correlated equilibria in extensive-form games, such as: “No-Regret Learning Dynamics for Extensive-Form Correlated Equilibrium” (NeurIPS 2020), “Efficient Deviation Types and Learning for Hindsight Rationality in Extensive-Form Games” (ICML 2021), and “Hindsight and Sequential Rationality of Correlated Play” (AAAI 2021), and others.

	(*) Line 95: Remove “player”.

	(*) Line 106: “We also will” is “We will also”.

	(*) Adding an example of extended game would help.

	(*) The terms used in the caption of Table 1 do not appear in the table.


SIGNIFICANCE

(+) The results could be of interest for the part of the NeurIPS community interested in algorithmic game theory.

(-) I have some concerns of the significance of the experimental results. It seems to me that the linear programming formulation does not scale well on large games, especially that one for communication equilibrium (which I think is the most interesting solution concept).

---

> ### Author Response · Authors · 2022-08-02
> **Response to review**
>
> Thanks for your review!
>
> We thank the reviewer for the brief suggestions in the box, and we will incorporate them into the final version.
>
> **The techniques employed by the authors are fairly standard, since they consist in a simple linear programming formulation.**
>
> We do not agree that the fact that we are using linear programming implies that our techniques are so “standard”. See the response to Reviewer TwPw for more specific comments.
>
> **I think that more formalism in proving the technical results is needed. There are only two propositions in the paper, whose proofs given in the appendix are not sufficiently formalized in mathematical terms. Moreover, I think a proof is also need in order to shows the correctness of the linear programming formulation.**
>
> If the reviewer is concerned about any particular part of the proofs, we will be happy to incorporate suggestions into the final version. See also the response to Reviewer TwPw regarding the specific kind of dualization referenced in the LP formulation.
>
> **I have some concerns of the significance of the experimental results. It seems to me that the linear programming formulation does not scale well on large games, especially that one for communication equilibrium (which I think is the most interesting solution concept).**
>
> The size of the LP formulation for full-certification equilibria scales effectively linearly with the game size (see Section 3.2). For communication equilibria, the scaling is quadratic (see end of Section 3.1). These scalings are reflected in the experiments. Both are polynomial in the size of the game. As we mention throughout the paper, this is better than algorithms for optimal correlated equilibria, which must scale exponentially (under reasonable complexity assumptions) due to NP-hardness results.
>
> We leave to future research the problem of scaling to even larger games, such as those where even enumerating the terminal nodes is infeasible.
>
> **Can you provide more details as to why you can restrict the space of messages by using the revelation principle (see footnote 3)? I think more details on this are need in the paper.**
>
> Footnote 3: the revelation principle states that any equilibrium can be transformed into one in which, in equilibrium, all players honestly report their true information and obey mediator recommendations. This assigns semantic meaning to the messages: the message called $I$ is the one sent by an honest player upon reaching infoset $I$, and the message called $a$ is the one that, when sent by the mediator, causes an honest player to play action $a$. Without the revelation principle, the messages would be simply arbitrary symbols with no semantic meaning that happen to be named the same as the actions and infosets.
>
> The revelation principle (Prop 3.4) ensures that every S-certification equilibrium can be expressed in $\hat\Gamma$, and its “converse” (Prop 3.5) ensures that every equilibrium that we discover in $\hat\Gamma$ (i.e., every solution to the LP (1)) is actually an S-certification equilibrium. Together, they ensure that we can restrict our attention to $\hat\Gamma$ without affecting the correctness or the optimality of the resulting program.
>
> **It is not clear to me what you mean by “truthful EFCE” and other “truthful” concepts. Can you provide some more details on this.**
>
> In truthful EFCE, the players must be incentivized to truthfully report their information to the mediator--they are allowed to lie. In regular EFCE, the mediator already knows the players’ information.

---

> > ### Comment · Reviewer_Xqpy · 2022-08-08
> > **Thanks for your response**
> >
> > I would like to thank the authors for their detailed response to my concerns. After reading other reviews, I still think that the paper is somehow weak in terms of technical contribution/novelty. Thus, I will stick to my score (borderline accept). Nevertheless, I am open to engage in a discussion with other reviewers, especially those that are promoting paper acceptance, and I will be happy to increase my score if they bring to me good reasons why the paper should be accepted.

---

### Official Review · Reviewer_ZsGj · 2022-07-11

**Rating:** 6
**Confidence:** 4
**Soundness:** 4 excellent
**Presentation:** 2 fair
**Contribution:** 3 good

**Summary:**

Many papers were dedicated to the study of correlated equilibrium in sequential games in recent years. In this work, the authors argue that focusing on (variants of) correlated equilibrium is too restrictive, as the inherent NP-hardness of finding optimal equilibria precludes them from being used in real-world applications. Because the root of the problem appears to be the imperfect recall of the mediator, i.e. the correlation device, they suggest studying a more general class of mediators, capable of not just sending messages to players, but also receiving them, effectively broadcasting some information. The main result presented in this paper claims that under the assumption that a certain property of signal sets holds, computing an optimal certification equilibrium in sequential games is a polynomial-time problem. As an intermediate result the authors obtain also an analogue of the revelation principle in their setting. Next, the authors introduce two special cases of the certification equilibria, allowing for faster computation (in case of the full-certification equilibria) or incorporation of the mediator’s own observations into their decision-making process (in case of the (S,M)-certification equilibrium). After a short section dedicated to a construction of the entire family of solution concepts based on the mediator’s recall and observations and the players’ abilities to lie and deviate, recovering several correlated equilibria as special cases, the authors move to the empirical evaluation. The main outcome of the empirical evaluation appears to be that when the mediator seeks to maximize social welfare, in the three classes of games examined by the authors, the algorithm computing the certification equilibria provides higher-quality solutions (in general) in a shorter time than the correlated equilibria.

**Questions:**

To strengthen the intuition about the presented concepts, could the authors give an example of some related setting (besides the case of imperfect recall of the mediator) that could NOT be modeled using a polynomially-computable certification equilibrium, e.g., because the NRC does not hold?

How robust are the results with respect to the optimized function? For example, do the certification equilibria perform well even when the mediator optimizes their own (let’s say) random utility?

How large are the games used in the experiments, i.e. with how many information sets?

How do the algorithms scale with the number of players? Could, e.g., the full-certification equilibrium be computed even in some moderately-sized game with four or five players?


**Limitations:**

I could not think of any other limitation not addressed already by the authors.


**Strengths And Weaknesses:**

The idea this paper examines seems extremely intriguing to me. The motivation is clear, and the connections the authors make between different solution concepts and the argumentation about their application validity and computability presents an interesting point of view which I am inclined to mostly agree with. As far as I can tell, after going through the appendix as well, the results seem to be original and correct. The experiments further corroborate the claims, letting the reader compare both the optimal values and the computational times, and the way how the authors visualize the results is coherent and easily comprehensible.

My main concern relates to this work’s overall exposition, though. The field of algorithmic game theory, and sequential games in particular, often requires a careful buildup, patiently guiding the reader through all the introduced concepts, because of their inherent complexity and an abundance of notation. Especially when discussing multiple types of equilibria. The authors certainly do not seem to be at fault here, but NeurIPS’ space limit does not appear to provide enough space for it. Because of that, even as someone fairly acquainted with the field, I had to reread several parts multiple times to understand what the authors are perhaps trying to say and how do the individual solution concepts they discuss relate. The absence of any examples besides the briefly described Fig1 further confuses the reader. For example, it took me a while to understand even why Theorem 3.2 immediately implies the polynomial computability of optimal communication equilibrium when it speaks about certification equilibria. Or, why program (1) is indeed linear. If I understand it well the players’ optimal strategies are basically projections from the mediator’s pure strategy space, is that correct? Many details are missing also from the experimental section, e.g., not explaining even informally which domains were employed and why mediator concepts make sense in those situations, literally referring the reader not even to the appendix, but to a completely different work.

To sum up, I feel like this work puts me in a difficult position: on one hand, I enjoyed the ideas the authors are presenting, but on the other it may be more suitable for a conference with a more generous space limit, or some journal. I vote to accept it as I believe that more examples and discussions could be put in the appendix (even though it is not optimal), and I look forward to further discussing these issues with the authors and fellow reviewers in the next reviewing phase.

---

> ### Author Response · Authors · 2022-08-02
> **Response to review**
>
> Thanks for your review!
>
> **Confusion on Thm 3.2 and Program (1)**
>
> Thm 3.2 applies to communication equilibria because communication equilibria are simply the special case of S-certification equilibria when S is unrestricted. We include more details about the LP construction in the response to TwPw above.
>
> We will use the extra page available in the final version to explicitly clarify the above points. There is also a small example in Appendix B, used to remark that certification equilibria can Pareto-dominate even normal-form coarse-correlated equilibria.
>
> **If I understand it well the players’ optimal strategies are basically projections from the mediator’s pure strategy space**
>
> We are not entirely sure what you mean here. The players’ best responses are (WLOG, by revelation principle) the honest strategies: report honest information and obey recommendations.
>
> **Details in experiments**
>
> Here is a brief description of each game family represented in Table 2. The games correspond to various applications of mediators in games that we discussed in the introduction of the paper.
>
> **B** is a two-player Battleship game, made into a nonzero-sum game by having each player value their own ships greater than their opponents’. The three numbers afterward are, in order, the length, width, and number of shots per side. Each player has a single ship of unit size.
>
> **S** is a simplified Sheriff of Nottingham game, a small game modeling a negotiation between two players. The first two numbers roughly correspond to how much power each player has in negotiation; the final number is the number of rounds of negotiation.
>
> **RS** is a small ridesharing game, played on a graph, in which the players walk around the graph attempting to serve “customers” who reside at certain nodes. The first number (1 or 2) denotes the specific graph on which the game is played; the second number is the number of steps in the game.
>
> We’ll add these descriptions to the final version.
>
> **Settings that cannot be solved in polynomial time, and NRC**
>
> If players cannot send messages to the mediator at all, and the mediator has no other way of gaining any information, we recover the notion of autonomous correlated equilibrium (ACE). It is NP-hard to compute optimal ACE, even in Bayesian games (see e.g. [24]).
>
> If NRC does not hold, one can still write down and solve the program (1) as it is defined in the paper, and it is still guaranteed to be an S-certification equilibrium by Prop 3.5 (the proof of that result does not actually require NRC: the inclusion of NRC as an assumption was an oversight in the submitted version, which we will fix in the final version). However, it is not guaranteed to be optimal.
>
> For an example, see the proof of Theorem 1 of [13]: that proof gives an instance in which, without NRC, there can be a social choice function (for us, an outcome distribution) that is not implementable by a direct mechanism (for us, mediator); since our program (1) explicitly restricts attention to direct mediators, it will fail to find such a distribution.
>
> The counterexample does not preclude the possibility of efficient algorithms for finding optimal S-certification equilibria without NRC, but we hope it gives intuition for why NRC is crucial to our construction. We leave open for future research the question of whether efficient algorithms exist for computing optimal S-certification equilibria without NRC.
>
> We will include this discussion explicitly in the final version.
>
> **How robust are the results with respect to the optimized function?**
>
> We are not entirely sure what you are asking. If you are asking whether our method supports utility functions other than the social welfare, it supports any von Neumann-Morgenstern utility function over the terminal nodes of the mediator-augmented game; for example, risk aversion, post-game payments by the mediator, etc. can be easily incorporated.
>
> If you are asking about the social welfare of an optimal equilibrium when the objective is not the social welfare: if the mediator has a random utility function, there is no reason for the optimal equilibrium to be any particular point: for example, if the mediator’s utility function is a random linear combination of the players’ utilities, we would obtain a random equilibrium on the boundary of the feasible payoff space. Whether such an equilibrium is likely to have good social welfare then depends on how large the payoff space is, which varies widely game as can be seen in Figure 1.
>
> **Sizes of games in experiments**
>
> The number of terminal nodes in each game is included in Table 2. As mentioned above, these games and sizes are standard in the literature on computing correlated equilibria.
>
> **Scaling with number of players**
>
> Our bounds depend only polynomially on the number of players, nodes, and sequences in the game–see the end of Sec 3.1 and beginning of Sec 3.2. So, yes: our algorithm easily scales to (reasonably-sized) games with many players.

---

### Official Review · Reviewer_TwPw · 2022-07-16

**Rating:** 4
**Confidence:** 4
**Soundness:** 3 good
**Presentation:** 2 fair
**Contribution:** 2 fair

**Summary:**

This paper studies the complexity of correlated equilibrium in extensive form games with mediator communication, knowns as the communication or certification equilibria. The main result is a polynomial-time algorithm, specifically a linear program formulation, for computing optimal communication and certification equilibria. To prove this result, the paper define a mediator-augmented games, which basically includes the mediator as an additional player. The new game has  polynomial size and can describe several equilibrium notions.


**Questions:**

"The infinite family of constraints can be made polynomial-sized by taking a dual of the inner optimization" at the end of Section 3.1 --- would you elaborate on how to achieve this, or is this a known result?



**Strengths And Weaknesses:**

Strengths:

-- the study of communication in extensive-form game seems an interesting and well-motivated problem.

-- the fact that for most correlation equilibrium notions, the optimal equilibrium is hard whereas this paper identifies polynomial time algorithm for CE with mediator is interesting.

Weakness:

-- The paper is very notation heavy without much illusion about the intuition of the model. For instance, I was hoping the paper could instantiate their model in one of the listed three applications (i.e., Crowdsourcing and ridesharing, persuasion or automated MD), to familiarize the audience with the setup and the motivation of the mediator.

-- the techniques of the work appears somewhat standard. It is not clear what the novelty is. Frankly speaking, I spent more time to understand the notations in Section 2 than to understand the proof of the main theorem, which is a natural conversion of the original game to a Stackelberg game with the mediator as the leader.

-- This is purely an equilibrium optimization paper without any aspect of learning. So it does not seem a good fit to NeurIPS.

---

> ### Author Response · Authors · 2022-08-02
> **Response to review**
>
> Thanks for your review!
>
> **I was hoping the paper could instantiate their model in one of the listed three applications (i.e., Crowdsourcing and ridesharing, persuasion or automated MD)**
>
> Our experiments contain some small instances of a ridesharing game (RS* in the experiments table; rightmost plot in Figure 1). In Appendix D, we more explicitly discuss the application to automated mechanism design. In the final version, we will use the additional page to add further intuition about the model. As to the motivation of a mediator for games, that is a common tool to define notions of equilibrium; cf. the various papers we cite on correlation and communication in games.
>
> **It is not clear what the novelty is. Frankly speaking, I spent more time to understand the notations in Section 2 than to understand the proof of the main theorem, which is a natural conversion of the original game to a Stackelberg game with the mediator as the leader.**
>
> While the use of LP to solve 2-player Stackelberg games is standard, the ability to apply the same techniques to our mediator-augmented game (which is general-sum with arbitrarily many players) is only a consequence of the revelation principle (finding Stackelberg equilibria in general-sum games is hard). Further, the construction of the mediator-augmented game itself includes steps that are fundamental to making the proof work and, to our knowledge, are novel. Concretely, the statement of revelation principle that we employ and its “converse” (Props 3.4 and 3.5), are new, and their careful statements are critical to the correctness of the game formulation and algorithm. In particular, the “usual” revelation principle of e.g. Forges [1986] is not sufficient to imply a polynomial-time algorithm, since the augmented game that it implies would have exponential size.
>
> In any case, we believe that the results we show in this paper reveal rather fundamental properties of how the presence of a mediator interacts with complexity of equilibrium computation–for example, that the key ingredient to efficient algorithms is the perfect recall of the mediator. Ironically, prior work on correlation in games has implicitly assumed an imperfect-recall mediator, which leads to worse mediation and, as we show, is also the source of the exponential complexity.
>
> **This is purely an equilibrium optimization paper without any aspect of learning. So it does not seem a good fit to NeurIPS.**
>
> NeurIPS has been the publication venue of numerous papers on equilibrium computation over the years that do not have “learning aspects”. In fact, NeurIPS best paper awards have been given to such papers. The call for papers explicitly welcomes papers on algorithmic game theory.
>
> **"The infinite family of constraints can be made polynomial-sized by taking a dual of the inner optimization" at the end of Section 3.1 --- would you elaborate on how to achieve this, or is this a known result?**
>
> Program (1) is a problem of the form
> $$\let\b\boldsymbol \max_{\b x \ge \b 0, \b B \b x = \b b} \b u^\top \b x~~~~~\text{s.t.}~~~~~\min_{\b y_i \ge \b 0, \b C_i \b y_i  = \b c_i} \b x^\top \b A_i \b y_i \ge \b v_i^\top \b x~~ \forall i \in [n]$$
> In this case,  $\b x = \b{\hat x}\_{\sf M}$ and $\b y_i = \b{\hat x}\_i$.  The other quantities appearing in the program ($\b{\hat x}^*, \b{\hat u}, \b{\hat z}$) are all constants. Taking duals of the inner minimizations results in a linear program, namely
> $$\max_{\b x \ge \b 0, \b B\b x = \b b} \b u^\top \b x~~~~~\text{s.t.}~~~~~\b C_i^\top \b z_i \le \b A_i^\top \b x,~~~\b c_i^\top \b z_i \ge \b v_i^\top \b x~~~~\forall i \in [n].$$
> This sort of dualization in optimization problems is a fairly standard approach; see e.g. [8] for the same approach applied to correlation, or [16] for a similar approach applied to zero-sum games.
>
> In the final version, we will elaborate on this and include a full description of the resulting program in the appendix.

---

### Meta-Review · Area_Chair_3n9p · 2022-08-26

**Recommendation:** Accept
**Confidence:** Certain

**Metareview:**

Executive summary:

Motivated by the potential hardness of computing optimal corrrelated equilibria, this paper looks at variants of correlated equilibria -- communication and certification equilibria --- where the mediator has additional power. The main result of the paper is a poly-time algorithm for computing optimal such equilibria, which embeds the mediator into the original game in the fashion of a two-player Stackelberg game and shows that the resulting game can be solved optimally via linear programming. The paper implies certain existing poly-time algorithm as special cases.

Discussion:

Overall this paper struck me as rather original, and the main result is rather general and encouraging. I think it may motivate follow-up motivation, and variants of it may even have practical use cases.

Accept.

**Award:**

No

---

### Decision · Program_Chairs · 2022-09-14

Accept